# Enhancing Neural Decompilation with Code-aware Fine-tuning and Inference-time Refinement

## Abstract

Language models (LMs) hold promise for automating binary decompilation by translating low-level assembly into high-level source code, helping code security analysis. However, current LM-based methods struggle with the complex control and data flows in real-world programs. We present TunerDEC, a Code-structure-aware LM-based approach to binary decompilation. TunerDEC combines a task-specific LM, fine-tuned on domain-specific data, with a general-purpose LM trained on generic corpora. The task-specific LM uses abstract syntax tree (AST) analysis to assign higher loss weights to key language constructs, such as loops, conditionals, and pointers, prioritizing their importance during training. At inference time, TunerDEC employs an iterative self-refinement process guided by compiler feedback and test-driven prompts, leveraging the general-purpose LM to improve the decompilation output. We evaluate TunerDEC on the HumanEval-Decompile dataset. The results show significant improvements in code readability, functional correctness, and robustness compared to state-of-the-art neural decompilation methods and an industry-strength decompiler.

## 1 Introduction

Decompilation translates low-level instructions back into a higher-level language. It can be found use in software reverse engineering (Faingnaert *et al.*, 2024), malware analysis (Mauthe *et al.*, 2021), and legacy code maintenance (Nitin *et al.*, 2021). However, decompilation is challenging as high-level programming constructs, such as variable names, loops, and compound data structures, are often stripped away during compilation. This results in low-level representations that obscure the original program's logic and intent, making accurate reconstruction complex and error-prone.

Progress for decompilation has been made through control-flow analysis (Verbeek *et al.*, 2020) and data-flow optimization (Zhang *et al.*, 2024), as well as the integration of machine learning models that outperform rule-based approaches (Hosseini and Dolan-Gavitt, 2022). Neural architectures, such as CodeBERT (Feng *et al.*, 2020), trained on code repositories, have demonstrated promising results in decompilation. These models can infer higher-level abstractions, rename variables, and rebuild source-level constructs to help human understanding. This shift towards machine learning has been further augmented by using richer intermediate representations and improved symbolic execution strategies to handle complex binaries. Representative neural-based solutions like SLADE (Armengol-Estapé *et al.*, 2024) and LLM4Decompile (Tan *et al.*, 2024) have shown promising results in code reconstruction. General-purpose language models (LMs), including DeepSeek-V3 (DeepSeek, 2024) and ChatGPT-4o (OpenAI, 2023), have also shown potential in code synthesis and transformation tasks through carefully designed prompt engineering, showing the potential of large, general-purpose LMs.

Neural-based decompilers, while promising, still face significant challenges in processing code compiled with aggressive optimization levels, such as `-O2` and `-O3`, which developers commonly use for compilation. Advanced code transformations, like function inlining, loop splitting, code merging, and instruction reordering, further obscure the original code structure, complicating the decompilation process. Language models often struggle with complex control flows, resulting in translated code that may fail to compile, deviate semantically from the original program, or lack the readability and characteristics of human-written code. Moreover, general-purpose models like GPT-4o, though powerful for a wide range of tasks, are not specifically trained on decompilation datasets and

lack the domain knowledge required for effective binary-to-source translation. As a result, ensuring syntactic correctness and functional equivalence in decompiled code remains an open challenge.

We present TUNERDEC, a neural framework for end-to-end binary-to-C decompilation. TUNERDEC leverages a fine-tuned, task-specific LM for translating assembly code into an initial C program and a general-purpose LM for refining the decompiled code. The task-specific LM (or decompiler), fine-tuned (using CodeLlama in this work) on domain-specific samples, is designed to capture key programming constructs like loops and recursion. During inference, the general-purpose LM (DeepSeek V3 and DeepSeek R1 in this work) refines the initial program using compiler feedback and test-driven prompts. This design combines the domain expertise of task-specific models with the generalization ability of large LMs through an iterative refinement process. As demonstrated in this paper, TUNERDEC improves functionality and structural fidelity across varying compiler optimization levels.

To fine-tune the task-specific LM as the decompiler, we introduce a *code-aware weighted token scheme* to support the underlying LM in capturing syntactic hierarchy and key constructs of the programming language. Each ground-truth training sample is parsed into an abstract syntax tree (AST), assigning higher loss weights to critical tokens like control-flow keywords and structural delimiters. This approach prioritizes essential structural elements, enabling the model to better preserve logical and syntactic integrity, even under aggressive compiler optimizations that obscure semantic cues.

During inference, TUNERDEC incorporates an *iterative refinement loop* using compiler feedback and test-driven prompts. If the decompiled output fails to compile or pass the automatically generated or user-provided test suite, a tailored prompt for a general-purpose LM is generated using the decompiled code, error logs, and the source assembly instructions from the binary. This iterative feedback mechanism corrects errors and progressively enhances the output, ensuring it closely aligns with the observed behavior of the original program.

We implement TUNERDEC by fine-tuning a 7B CodeLlama model as the task-specific decompiler and using DeepSeek-V3 as a general-purpose LM for test-time refinement. We compare TUNERDEC against five baselines: Ghidra (Agency, 2024), an industry-strength rule-based decompiler, and four LLM-based methods: ChatGPT 4o (OpenAI, 2023), DeepSeek-V3 (DeepSeek, 2024), LLM4Decompile-End-6.7B (Tan *et al.*, 2024), and Codellama-7b-Instruction (Roziere *et al.*, 2023). We test all methods on the HumanEval-Decompile benchmark (adapted from HumanEval (Chen *et al.*, 2021)) using binaries compiled with four GCC optimization levels (O0 - O3). We consider four evaluation metrics: re-compilation success, re-execution correctness, edit similarity, and control flow reconstruction accuracy to the original C code. TUNERDEC consistently outperforms all baselines, generating more readable and semantically accurate C code.

This paper makes the following contributions:

- It presents a new approach to combine task-specific and general-purpose LMs for decompilation;
- It introduces a code-aware fine-tuning technique to improve the ability of LMs in reasoning code structures;
- It presents an iterative refinement loop to integrate compiler feedback and test-driven prompts to improve the semantic correctness of the decompiled output.

## 2 MOTIVATION

Figure 1 shows a motivational example of translating the assembly code of a Fibonacci program back into C, which is not in the fine-tuning dataset of TUNERDEC. Figure 1(a) shows the source, recursive implementation of the Fibonacci sequence. For this example, we compiled the source code using `gcc -O3` to generate an x86 executable binary, which was then disassembled using `objdump` to produce the assembly code. We removed the code bytes and platform-specific information, retaining only the address and function-related assembly code, which was then fed into different decompilers, and the outputs were evaluated.

### 2.1 RESULTS

We tested Ghidra (Ver.    11.3.1), an industry-strength decompiler, two general-purpose LLMs, DeepSeek-V3 (Release on March 2025) and ChatGPT 4o, a neural-based decompiler, LLM4Decompile alongside TUNERDEC, and a variant of TUNERDEC. To verify the correctness of the generated code, we leverage DeepSeek-V3 to generate test inputs from the source C code. We then feed the input to the source binary and use the output of the binary execution as the test oracle.

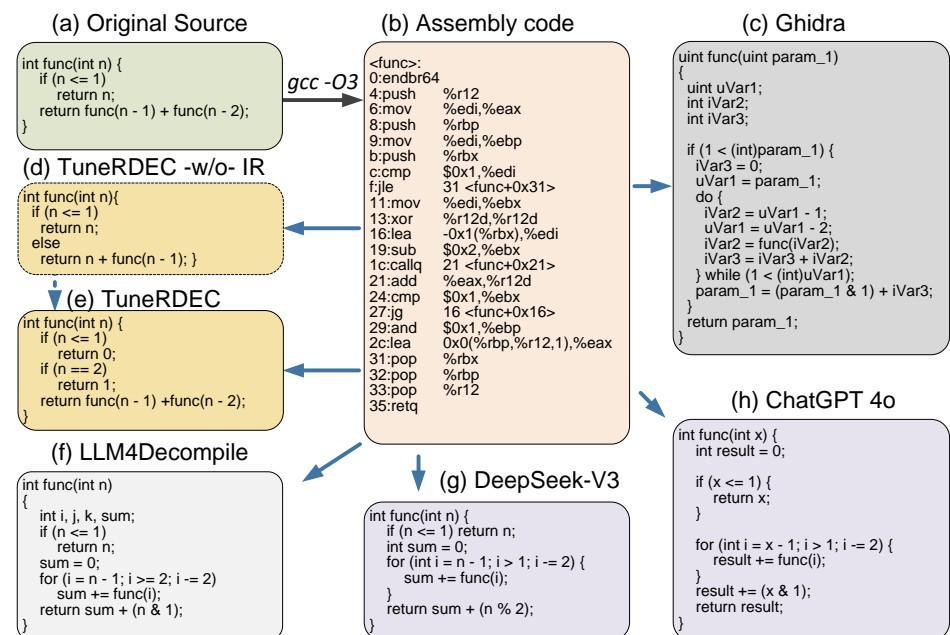

Figure 1: Decompilation results of various approaches (c - h) to the ground truth (a).

**Ghidra** in Figure 1c introduces unnecessary complexity through the use of a loop and multiple intermedate variables (e.g., `uVar1`, `iVar2`). The control flow of the Ghidra-generated code is more complex than the original, straightforward recursive logic, making it harder to read and understand. While rule-based decompilers like Ghidra, Hex-Rays (Hex-Rays, 2024), and RetDec (Křoustek *et al.*, 2017) can generate functionally correct code that passes test cases, the resulting code is often difficult to interpret, especially for aggressively optimized binaries. This makes it difficult for developers to conduct code inspections. In light of these observations, this paper focuses on neural-based decompilation using LMs.

**DeepSeek-V3** in Figure 1g recognizes a recursive pattern and attempts to reconstruct a loop. However, the inferred recursion does not follow the standard Fibonacci recurrence. Instead, it sums a subset of recursive calls in steps of two (`i -= 2`), plus an odd/even offset. As a result, its translation is not equivalent to the source code.

**ChatGPT 4o** in Figure 1h produces a syntactically correct C program that can compile and run. However, the function uses a similar "odd-step" recursion as Deepseek-V3 and LLM4decompile, rather than the standard `f(n-1) + f(n-2)` Fibonacci scheme implemented in the source code.

**LLM4Decompile** in Figure 1f also produces a C code that re-creates a recursive structure and recognizes the base case `if (n <= 1) return n;`, but fails to recover the correct recursive structure. This suggests it mimicked the syntax and partial behavior but lost the semantic essence.

**TUNERDEC-w/o-IR** in Figure 1d relies solely on a local code-aware LM-based decompiler, which prioritizes key control-flow structures like the if statement and the recursive function call. Although it demonstrates a strong understanding of control-flow patterns, it does not fully recover the correct logic of the Fibonacci sequence.

**TUNERDEC** in Figure 1e follows a two-stage pipeline. It first uses a decompiler to generate the initial C code and then refines the decompiled code (from TUNERDEC-w/o-IR) through the feedback of compilation and test case execution. In the second stage of TUNERDEC, it uses a generic LM and test execution results to successfully improve the decompiled code, leading to a translation that faithfully represents the source implementation.

## 2.2 INSIGHT

Widely used compiler optimizations like GCC's `-O3` can aggressively transform and obscure a program's structure, making source code reconstruction challenging. Our key insight is to guide LMs in mapping low-level instructions to high-level constructs such as loops, conditionals, and recur-

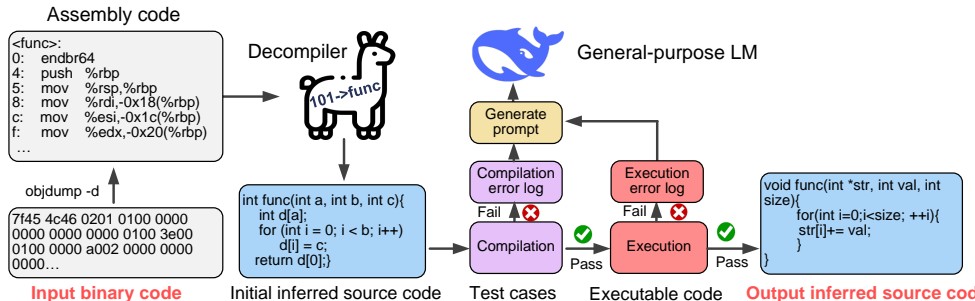

Figure 2: The workflow of TUNERDEC involves inputting binary code into a fine-tuned CodeLlama for initial decompilation. If the inferred code fails the test case, TUNERDEC refines it using error logs with the help of DeepSeek-V3/DeepSeek-R1, a general-purpose LM, for iterative refinement to improve the accuracy and quality of the decompiled output.

sion, enabling the generation of human-readable code. By incentivizing LMs to learn and restore these constructs, TUNERDEC produces high-quality, executable code that faithfully reproduces the program's behavior, even under aggressive optimizations. Additionally, TUNERDEC uses compiler feedback and test case execution to iteratively refine the code, identifying and correcting errors undetected by syntax analysis. This approach ensures accurate decompilation and enhances the quality of generated code.

## 3 OVERVIEW

Figure 2 illustrates the workflow of TUNERDEC. Starting with an executable binary, it uses `objdump` to disassemble it into assembly instructions. The assembly instructions are passed to the LLM-based decompiler to generate initial C code to be compiled using a standard compiler like GCC. If compilation fails or the resulting binary exhibits runtime errors, the system initiates an iterative refinement process, with a maximum of three total iterations. In each iteration, the assembly code, C code, and error logs are given to a generic LM (e.g., DeepSeek-V3) for iterative refinement. The generic LM updates the C code based on compilation or test run feedback, and the revised code is recompiled and executed with auto-generated test inputs. If the code still fails after the initial iterations with DeepSeek-V3, the refinement is escalated to a more capable model (e.g., DeepSeek-R1) with explicit chain-of-thought reasoning, which continues the process within the remaining iteration budget. The final C code is accepted if it compiles and executes successfully, otherwise, decompilation is deemed unsuccessful.

## 4 CODE-AWARE FINE-TUNING FOR DECOMPILATION

Figure 3 depicts the TUNERDEC fine-tuning process, which adapts Meta's pre-trained 7B CodeLlama model for the task-specific LM. However, our methods can be applied to other LMs as well. To prepare the fine-tuning training data, we compile C code into binaries with various compiler optimization levels (O0, O1, O2 and O3 in this work) to create a diverse dataset. These binaries are disassembled into assembly code using tools like `objdump`, and each code segment (e.g., individual functions or functions with callers and callees) is paired with its corresponding C code. The LM is trained to respond to a prompt asking to translate assembly code to C. The C code is also parsed into an AST to capture its syntactic hierarchy. AST nodes are tokenized, and key constructs (e.g., control-flow keywords) are weighted higher in the loss computation. The goal is for the model to accurately generate C source code from the corresponding assembly input.

### 4.1 TOKENIZATION

Let $\mathcal{D} = \{(X_i, Y_i)\}_{i=1}^{N}$ be the training dataset, where each $X_i = (x_{i,1}, \ldots, x_{i,L_i})$ represents the assembly code, and $Y_i = (y_{i,1}, \ldots, y_{i,M_i})$ is the corresponding C source code. Define $p_\theta(y_{i,m} \mid y_{i,<m}, X_i)$ as the probability of generating the $m$-th token $y_{i,m}$ in the C code, given all previously generated tokens $y_{i,<m} = (y_{i,1}, \ldots, y_{i,m-1})$ and the input assembly sequence $X_i$. Each training example $(X_i, Y_i)$ is processed in a sequence-to-sequence format. Specifically, the model receives a prompt:

```
Translate the input x86 assembly code into C code.
### Input: {Assembly code}
### Response: {Source C code}
```

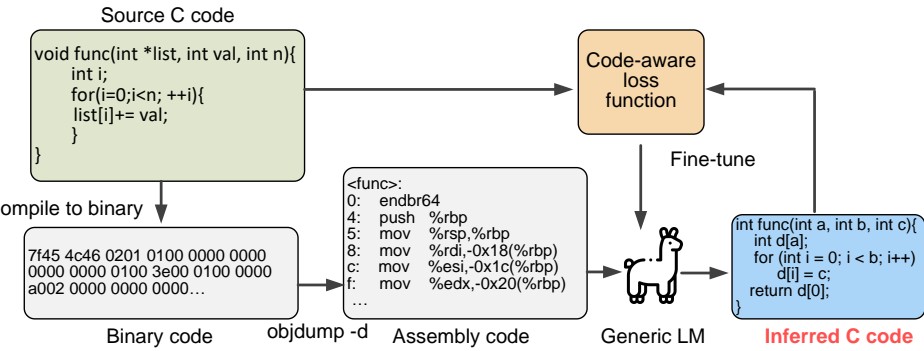

Figure 3: TUNERDEC's fine-tuning strategy.

Table 1: Token categorization in our code-aware weighted token scheme.

| Category | Examples | Rationale ($base\_weight$) |
|---|---|---|
| **Loops** | `for`, `while`, `do` | Preserving loop constructs is crucial for accurate control-flow reconstruction (2.0). |
| **Conditionals** | `if`, `else`, `switch`, `case` | Correctly capturing decision points is essential for semantic equivalence (1.8). |
| **Pointers** | `*`, `->`, `&` (when used as address operators) | Pointer usage can drastically affect memory access patterns and correctness (1.5). |
| **Control Flow** | `break`, `continue`, `goto` | These tokens can alter the natural flow of execution, requiring careful handling (1.6). |
| **Default** | All other tokens | Tokens not classified above remain at a baseline weight (1.0). |

The training process optimizes model parameters $\theta$ by minimizing the code-aware cross-entropy loss between the predicted tokens and the ground-truth tokens in $Y_i$. Tokenization is performed using the pre-trained CodeLlama-7B tokenizer, and input prompts are truncated to a maximum length of 4096 tokens.

## 4.2 CODE-AWARE WEIGHTED TOKEN SCHEME

To incorporate syntactic information from the AST, we introduce per-token weights $w_{i,m} \geq 1.0$, where higher weights are assigned to tokens corresponding to semantically important language constructs (e.g., loops, conditionals, or pointer-related operations). Specifically, we parse each code snippet into an AST using `pycparser` (Bendersky, 2023), which exposes the hierarchical structure of control flow, memory access patterns, struct field access, and other syntactic constructs. We then traverse the AST to identify all relevant constructs listed in Table 1, and compute their *nesting depth* within the tree. The weighting scheme combines domain knowledge with empirical tuning. We first initialize base weights according to programming-language intuition (e.g., loops and conditionals influence execution flow more than identifiers or arithmetic). We then apply a lightweight grid search on a held-out validation set. Specifically, candidate base weights for key constructs were sampled from ranges [1.0–2.5], with a step size of 0.1. The final configuration was chosen to maximize re-executability while preventing any single construct from dominating the loss. Finally, a logarithmic depth multiplier, $\log_2(1 + \text{depth}(t))$, was applied so that deeper constructs receive more emphasis in a controlled manner:

$$w_t = \text{base\_weight}(t) \cdot \log_2(1 + \text{depth}(t))$$

Here, $w_t$ denotes the final weight assigned to token $t$ and base_weight($t$) is a static weight determined by the construct type (e.g., `if`, `while`). Tokens not associated with any predefined construct are assigned a default baseline weight of 1.0. This weighting scheme rewards the correct generation of syntactically significant elements, guiding the model to focus more on recovering the underlying program logic during decompilation. We increase weights with AST nesting depth to reflect the greater structural importance of deeply embedded constructs. The resulting Code-aware cross-entropy loss is:

$$\mathcal{L}_{\text{weighted}}(\theta) \;=\; -\sum_{i=1}^{N} \sum_{m=1}^{M_i} w_{i,m} \, \log\Big( p_\theta\big(y_{i,m} \,|\, y_{i,<m},\, X_i\big)\Big).$$

$N$ is the number of training examples, $M_i$ is the length of the target sequence $Y_i$ for the $i$-th example. We seek the optimal model parameters $\theta^*$ by minimizing this weighted loss function:

$$\theta^* \;=\; \arg\min_\theta \mathcal{L}_{\text{weighted}}(\theta).$$

Table 2: Distribution (%) of code patterns.

| Pattern | Exebench-Train | HumanEval-Decompile |
|---|---|---|
| Loops | 52.6 | 91.5 |
| Conditionals | 47.7 | 83.5 |
| Pointers | 55.7 | 86.0 |
| Data Structures | 10.4 | 0.6 |
| Control Flow | 7.6 | 7.9 |

By applying larger weights to tokens that embody crucial language structures, this code-aware weighted token scheme encourages the model to prioritize syntactic fidelity and functional correctness in the generated source code.

### 4.3 TRAINING DATA GENERATION

To build the training dataset for fine-tuning the task-specific LM, we utilize the *train real compilable* dataset from ExeBench (Armengol-Estapé *et al.*, 2022), which comprises 660 K compilable C code snippets. To prepare the training data, we first generate hashes for both the `input` (assembly code) and `output` (C code) fields of each training sample. We precompute Term Frequency-Inverse Document Frequency (TF-IDF) vectors (Arroyo-Fernández *et al.*, 2019) for all input and output texts to quantify their textual similarity. We filter out highly similar code pairs by applying a similarity threshold of 0.8, reducing the dataset to 80 K unique files and minimizing redundancy to prevent overfitting during training. Following the deduplication step, each function within the dataset is compiled using `gcc -c` with optimization levels `-O0`, `-O1`, `-O2`, and `-O3` to remove any uncompilable code. The resulting binaries are then disassembled into assembly code using `objdump`, and we explicitly remove all code bytes and platform-specific information, retaining only the addresses and raw instruction mnemonics. Table 2 presents the distribution of different code constructs in the training samples. To enhance the dataset's quality and ensure consistency, we process the disassembled code by eliminating information that is redundant to decompilation, such as extraneous annotations and metadata, while retaining line numbers. Preserving line numbers is essential for maintaining the structural alignment between assembly instructions and their corresponding C code, facilitating accurate decompilation. This meticulous preprocessing culminated in a final dataset encompassing ∼243 million tokens.

### 4.4 FINE-TUNING METHOD

We use Low-Rank Adaptation (LoRA) (Hu *et al.*, 2021) with 8-bit quantization (Dettmers *et al.*, 2022) to fine-tune a 7B CodeLlama model as the decompiler. LoRA enhances model adaptability by introducing low-rank trainable matrices specifically within the query, key, value, and output projection modules ($q\_proj$, $k\_proj$, $v\_proj$, $o\_proj$) of the transformer architecture. By restricting LoRA modifications to these projection layers, we reduce the overall number of trainable parameters by approximately 0.2%, significantly lowering computational overhead and accelerating convergence during fine-tuning. Empirical studies (Nijkamp *et al.*, 2022; Li *et al.*, 2022) demonstrate that modifying attention projection layers significantly improves performance in various NLP tasks, validating our focus on these components for binary decompilation. We configure LoRA with a rank $r = 16$ and a scaling factor $\alpha = 16$, alongside a dropout rate of 5%, to ensure robust learning and prevent overfitting. Additionally, we integrate the custom code-aware weighted loss function into the fine-tuning process. This loss function assigns different token-level weights to emphasize critical tokens, such as key patterns and control structures in binary code. Specifically, we calculate a weighted cross-entropy loss by scaling each token's contribution with its corresponding weight. In this unnormalized form, sequences containing more high-importance constructs (e.g., loops, conditionals, pointers) exert proportionally greater influence during training. This approach prioritizes high-importance tokens, guiding LoRA to focus on the most significant aspects of binary decompilation while minimizing the influence of less important tokens.

## 5 INFERENCE-TIME REFINEMENT

To enhance the executability of decompiled binaries from the fine-tuned decompiler, TUNERDEC incorporates an iterative refinement step using the general-purpose LMs (DeepSeek-V3 and DeepSeek-R1 in this work). This framework corrects discrepancies and errors in the initial C source code, ensuring the final output is both syntactically correct and functionally equivalent to the original binary.

## 5.1 Iterative Refinement Process

The refinement process begins with the output from the decompiler, which generates an initial version of the C source code based on the input assembly code. This inferred code is then subjected to compilation and execution tests to verify its validity and functionality. If the code successfully compiles and executes without errors, it is accepted as the final decompiled output. However, in instances where the code fails to compile or execute correctly, we initiate an iterative refinement process using the generic LMs to correct the errors and improve the code quality.

## 5.2 Prompt Construction and Instructions

To facilitate effective communication with our general-purpose LM, we construct a prompt encompassing all necessary information for error resolution. The prompt includes the original assembly code, the inferred C source code, the test case, and the observed error logs from the failed compilation or execution attempts. Additionally, we provide structured instructions to guide the general-purpose LM in refining the code (see Appendix **??**). If the refined code still fails to compile or execute correctly, the new error logs are incorporated into an updated prompt, and the general-purpose LM is consulted again for further corrections. To balance thoroughness with computational efficiency, this refinement cycle is limited to three iterations. If the code fails to pass the tests after three attempts, the decompilation process is deemed unsuccessful for that particular binary.

## 6 Evaluation Setup

### 6.1 Platforms and Evaluation Datasets

Our experiments are conducted on an NVIDIA A800 GPU with 80 GB of GPU memory. We use a fine-tuned 8-bit quantized CodeLlama 7B model as the decompiler and DeepSeek-V3/R1 as the general-purpose LM. We apply TUNERDEC and the baseline methods to HumanEval-Decompile, which consists of 164 C programs adapted from the Python-based HumanEval dataset. It is designed to assess the re-executability of decompilation systems and is used to evaluate LLM4Decompile Tan *et al.* (2024). We also curate a 378-problem LeetCode benchmark by scraping publicly available statements, I/O specifications, and reference solutions. We filter for deterministic I/O, remove multi-module or interactive tasks, deduplicate near-identical solutions, and normalize each problem to a single C entry point, yielding a function-level suite that stresses arithmetic, strings, arrays, hash maps, and dynamic programming across four optimization levels. We compile each C program using four GCC optimization flags (-O[0-3]). We exclude all code snippets that overlap with our test benchmarks from the ExeBench training set.

### 6.2 Competitive Baselines

We compare TUNERDEC against five baselines. These include four neural decompilers: DeepSeek V3, ChatGPT 4o, CodeLlama-7B-Instruct (Roziere *et al.*, 2023), and LLM4Decompile-End-6.7B (Tan *et al.*, 2024), and Ghidra (Version 11.3.1), a state-of-the-art rule-based decompiler. All LMs are evaluated using the same prompts for fairness. We select the 6.7B version of LLM4Decompile, whose parameter size is comparable to our 7B decompiler.

### 6.3 Evaluation Methodology

We evaluate our approach on four metrics: *Re-compilability*, *Re-executability*, *Edit Similarity*, and *CFG full match accuracy*. Re-compilability measures the percentage of decompiled code snippets that compile successfully without errors, indicating correct syntax and structure for the C compiler. Re-executability assesses the percentage of decompiled code that not only compiles but also passes the original test cases, ensuring functional correctness and behavioral fidelity to the original binary. Edit Similarity, as used in (Armengol-Estapé *et al.*, 2024), evaluates how closely the decompiled code matches the ground-truth source code. We also evaluate the full match accuracy of the CFG of the decompiled source code by comparing it to the CFG of the ground truth source code. To assess *CFG full match accuracy*, we use the `isomorphic(G1, G2)` function from the Python `networkx` library, which determines whether two graphs are structurally identical. Specifically, it checks whether a one-to-one mapping exists between the nodes of G1 and G2 that preserves all edges, implying that both graphs have the same number of nodes and edges, and their connectivity can be relabeled to match exactly. Finally, we investigate the impact of different optimization levels using four GCC options (-O0, -O1, -O2, and -O3), acknowledging that compiler optimizations can lead to varying assembly outputs.

Table 3: Re-compilability rates, re-executability rates, and edit-similarity of different approaches on the 164 × 4 HumanEval-Decompile benchmark. TUNERDEC w/o IR refers to the TUNERDEC results without using the Iterative Refinement scheme.

| Opt Flag | Re-compilability (%) | | | | | Re-executability(%) | | | | | Edit Similarity(%) | | | | |
|---|---|---|---|---|---|---|---|---|---|---|---|---|---|---|---|
| | -O0 | -O1 | -O2 | -O3 | Avg. | -O0 | -O1 | -O2 | -O3 | Avg. | -O0 | -O1 | -O2 | -O3 | Avg. |
| Ghidra | 23.7 | 20.7 | 16.4 | 17.6 | 19.6 | 19.5 | 14.6 | 12.2 | 11.5 | 14.4 | 30.9 | 28.2 | 28.1 | 26.9 | 28.5 |
| DeepSeek-V3 | 93.3 | 92.7 | 88.4 | 79.3 | 88.4 | 48.2 | 33.5 | 25.0 | 22.0 | 32.2 | 55.2 | 41.3 | 40.8 | 38.2 | 43.9 |
| ChatGPT 4o | 94.5 | 91.4 | 90.2 | 84.7 | 90.2 | 23.7 | 14.0 | 12.2 | 14.6 | 16.1 | 45.0 | 39.9 | 39.4 | 37.0 | 40.3 |
| Codellama-7b-Ins. | 62.2 | 60.9 | 59.7 | 56.1 | 59.7 | 1.8 | 1.2 | 1.8 | 1.2 | 1.5 | 35.1 | 35.0 | 34.2 | 33.2 | 34.4 |
| LLM4Decompile-End-6.7B | 93.2 | 88.4 | 91.4 | 88.4 | 90.4 | 69.5 | 43.9 | 40.9 | 36.6 | 47.7 | 71.9 | 54.2 | 53.0 | 50.6 | 57.4 |
| TUNERDEC-w/o-IR | 94.5 | 93.3 | 93.3 | 90.9 | 93.0 | 67.1 | 50.0 | 49.4 | 47.0 | 53.4 | 69.2 | 61.2 | 60.7 | 60.2 | 62.8 |
| DeepSeek-V3-w-IR | 96.3 | 94.5 | 93.2 | 92.0 | 94.0 | 78.7 | 65.2 | 65.2 | 58.5 | 66.9 | 69.6 | 49.0 | 48.9 | 44.5 | 47.5 |
| TUNERDEC (ChatGPT 4o) | 98.2 | 95.1 | 96.3 | 97.6 | 96.8 | 79.3 | 64.0 | 62.8 | 60.4 | 66.6 | 65.8 | 56.6 | 55.7 | 55.0 | 58.3 |
| TUNERDEC | **100** | **100** | **100** | **100** | **100** | **95.1** | **90.9** | **90.9** | **89.6** | **91.6** | **71.0** | **60.8** | **60.6** | **58.0** | **62.6** |

Table 4: Re-compilability rates, re-executability rates, and edit-similarity of different approaches on 378 × 4 LeetCode benchmark. TUNERDEC w/o IR refers to the TUNERDEC results without using the Iterative Refinement scheme.

| Opt Flag | Re-compilability (%) | | | | | Re-executability(%) | | | | | Edit Similarity(%) | | | | |
|---|---|---|---|---|---|---|---|---|---|---|---|---|---|---|---|
| | -O0 | -O1 | -O2 | -O3 | Avg. | -O0 | -O1 | -O2 | -O3 | Avg. | -O0 | -O1 | -O2 | -O3 | Avg. |
| Ghidra | 18.2 | 17.4 | 15.4 | 15.2 | 16.5 | 17.0 | 14.3 | 12.0 | 12.1 | 13.8 | 31.3 | 30.0 | 30.4 | 26.5 | 29.5 |
| DeepSeek-V3 | 80.2 | 72.6 | 69.1 | 66.5 | 72.1 | 46.7 | 31.7 | 32.1 | 29.6 | 35.0 | 60.0 | 48.3 | 47.5 | 45.2 | 50.2 |
| ChatGPT 4o | 56.8 | 53.7 | 51.0 | 47.9 | 52.3 | 19.8 | 14.7 | 12.7 | 10.5 | 14.4 | 50.3 | 45.0 | 44.7 | 41.8 | 45.4 |
| Codellama-7b-Ins. | 34.4 | 34.2 | 27.8 | 27.6 | 31.0 | 4.3 | 3.9 | 3.5 | 3.5 | 3.8 | 42.5 | 41.3 | 41.4 | 39.8 | 41.2 |
| LLM4Decompile-End-6.7B | 67.7 | 64.9 | 64.9 | 62.7 | 65.0 | 46.7 | 32.4 | 32.1 | 31.1 | 35.6 | 60.7 | 56.7 | 55.7 | 55.5 | 57.2 |
| TUNERDEC-w/o-IR | 69.9 | 67.2 | 66.7 | 65.3 | 67.3 | 50.1 | 35.5 | 34.6 | 33.3 | 38.4 | 63.7 | 60.2 | 59.8 | 57.7 | 60.3 |
| DeepSeek-V3-w-IR | 87.5 | 83.3 | 77.8 | 73.8 | 82.9 | 65.6 | 51.5 | 50.2 | 39.8 | 55.7 | 60.3 | 59.4 | 53.7 | 50.7 | 56.0 |
| TUNERDEC (ChatGPT 4o) | 84.1 | 78.8 | 78.6 | 78.0 | 80.5 | 69.7 | 64.9 | 63.0 | 62.9 | 65.9 | 56.1 | 51.7 | 51.0 | 50.4 | 52.3 |
| TUNERDEC | **89.9** | **90.7** | **90.4** | **89.5** | **90.1** | **72.0** | **71.0** | **73.4** | **69.7** | **71.5** | **62.4** | **69.2** | **59.5** | **58.3** | **62.3** |

## 7 EXPERIMENTAL RESULTS

Table 3 and 4 report the performance of different approaches on the HumanEval-Decompile and LeetCode benchmarks across compiler optimization levels and evaluation metrics. TUNERDEC-w/o-IR refers to the TUNERDEC without using generic LMs for Iterative Refinement (IR). Additionally, we evaluate DeepSeek-V3 with our iterative refinement scheme (termed DeepSeek-V3-w-IR). In terms of not leveraging IR, Ghidra shows the lowest performance across all metrics, this highlights the limitations of rule-based decompilers in handling optimized binaries, especially under higher optimization flags (-O2, -O3). For general-purpose LMs, ChatGPT 4o, DeepSeek-V3 show strong re-compilability (>70% in most cases) but much lower re-executability and edit similarity. These models often generate compilable but functionally incorrect code. DeepSeek-V3 outperforms ChatGPT 4o in re-executability and edit similarity. TUNERDEC reaches 100% re-compilability and 91.6% average re-executability, substantially exceeding all baselines on the HumanEval-Decompile; without refinement, TUNERDEC-w/o-IR already leads the non-IR group (93.0% / 53.4% / 62.8% on recomp/exe/edit), outperforming LLM4Decompile especially at higher optimization flags. On the larger and more heterogeneous LeetCode suite, TUNERDEC maintains 90.1% re-compilability and 71.5% re-executability on average, again topping DeepSeek-V3-w-IR. General LLMs (ChatGPT-4o, DeepSeek-V3) show high re-compilability but low re-executability/edit similarity, indicating compilable yet behaviorally incorrect code. Our code-aware fine-tuning yields drafts, IR, then applies small, diagnostics-guided edits, raising both compile and run correctness. DeepSeek-V3-w-IR improves over its base model but remains below TUNERDEC because it starts from less stable drafts and tends to over-edit.

Although LLM4Decompile-End performs relatively well at -O0 in terms of re-executability, likely due to its ability to handle simpler, less optimized assembly, its performance degrades significantly at higher optimization levels (-O1, -O2, -O3). At these levels, aggressive compiler optimizations transform control flow and code structure, making high-level semantics harder to recover. This poses challenges for LLM4Decompile, which struggles to preserve logical consistency in the decompiled output, unlike TUNERDEC-w/o-IR. Incorporating Iterative Refinement (IR) leads to significant performance gains. TUNERDEC with IR achieves perfect 100% re-compilability, 91.6% re-executability, and 62.6% edit similarity on HumanEval-Decompile benchmark. Specifically, 100% re-compilability confirms syntactic correctness, but the remaining 8.4% failures in re-executability, which stem from subtle semantic errors that cannot be caught by the compiler but fail during execution with test cases. These are often attributed to data-flow misalignments (such as confusing ' > ' with ' ≥ '). Compared to TUNERDEC-w/o-IR, this represents an absolute gain of 38.2% in re-executability (from 53.4% to 91.6%), highlighting the effectiveness of iterative refinement. Sim-

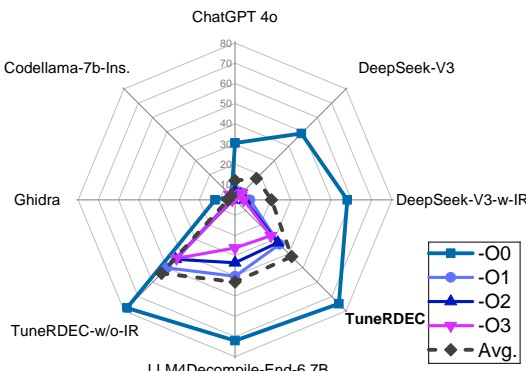

Figure 4: Performance in CFG reconstruction ability (Full match accuracy%) on the HumanEval-Decompile benchmark.

ilarly, DeepSeek-V3-w-IR shows a substantial re-executability gain of 34.7% over its base model, confirming the general effectiveness of iterative refinement in enhancing functional correctness.

We also integrate ChatGPT-4o as the backbone of TUNERDEC. Our evaluation shows that TUNERDEC (ChatGPT-4o) improves both re-compilability and re-executability on the HumanEval-Decompile and MDPP suites. However, it experiences a noticeable drop in edit similarity compared to TUNERDEC without Iterative Refinement (TUNERDEC-w/o-IR). This suggests that ChatGPT-4o, while enhancing functional correctness, tends to modify the initial predictions more aggressively. In contrast, TUNERDEC (DeepSeek) better preserves the original semantic structure, indicating that DeepSeek's generation capability is more aligned with the intent of the initial inference.

## 7.1 CODE-AWARE WEIGHTED LOSS VS. DEFAULT LOSS FUNCTION

To evaluate the impact of our code-aware weighted token loss, we conducted an ablation study comparing TUNERDEC 's fine-tuning strategy, which uses AST-guided token weighting, with a baseline using uniform cross-entropy loss. The TUNERDEC's code-aware loss emphasizes critical tokens (e.g., control-flow keywords, structural delimiters) based on their AST nesting depth to preserve program logic and structure. In contrast, the baseline treats all tokens equally. We fine-tune two CodeLlama-7B models using LoRA (rank $r = 10$, 5% dropout) on the Exebench benchmark and evaluate them on the HumanEval-Decompile benchmark, which includes 164 C programs compiled with GCC optimizations (-O0 to -O3). The only difference between models was the loss function: one used our weighted scheme; the other used a uniform weight of 1.0. Both are evaluated without iterative refinement to isolate the effect of the loss.

Table 5 shows that the code-aware weighted loss consistently outperforms the uniform loss across all metrics, with the most significant improvements in re-executability (a 9.7% average improvement) and edit similarity (an 8.5% average improvement).

The superior performance of the code-aware weighted loss function demonstrates its effectiveness in enhancing the fine-tuned 7B CodeLlama model's ability to produce accurate initial decompilation outputs. By prioritizing critical syntactic constructs (e.g., control-flow keywords, structural delimiters), the weighted loss improves the reconstruction of program logic and semantic fidelity, particularly for binaries with aggressive optimizations. For instance, at -O3, the code-aware loss improves re-executability by 8.6% (47.0% vs. 38.4%) and edit similarity by 8.1% (60.2% vs. 52.1%), reflecting its ability to handle complex transformations like loop splitting and function inlining. The CFG full match accuracy also improves significantly (7.5% average, up to 10.2% at -O0), indicating better preservation of structural fidelity.

## 7.2 ANALYSIS OF THE CFG RECONSTRUCTION ABILITY OF DIFFERENT DECOMPILATION APPROACHES

Figure 4 presents the full match accuracy of CFG construction for different decompilation approaches across compiler optimization levels. Ghidra performs poorly across all optimization levels, with an average of only 5.9%, indicating its limited ability to reconstruct high-level control structures. DeepSeek-V3 achieves 17.4% average full match accuracy, which is better than ChatGPT 4o (12.0%) and CodeLlama-7b-Instruction (5.2%). LLM4Decompile-End significantly improves CFG reconstruction ability over general-purpose LMs, achieving an average full match accuracy of

Table 5: Performance comparison of loss functions on the HumanEval-Decompile benchmark across GCC optimization levels (-O0 to -O3).

| Loss Function | Opt. Level | Re-compilability (%) | Re-executability (%) | Edit Similarity (%) | CFG Full Match Accuracy (%) |
|---|---|---|---|---|---|
| Default Loss (cross-entropy) | -O0 | 91.6 | 54.2 | 59.3 | 67.8 |
| | -O1 | 89.5 | 41.4 | 53.3 | 48.8 |
| | -O2 | 88.8 | 40.8 | 52.4 | 36.5 |
| | -O3 | 86.7 | 38.4 | 52.1 | 31.8 |
| | Avg. | 89.1 | 43.7 | 54.3 | 46.3 |
| Code-Aware Weighted Loss | -O0 | 94.5 | 67.1 | 69.2 | 78.0 |
| | -O1 | 93.3 | 50.0 | 61.2 | 50.0 |
| | -O2 | 93.3 | 49.4 | 60.7 | 43.9 |
| | -O3 | 90.9 | 47.0 | 60.2 | 43.3 |
| | Avg. | 93.0 | 53.4 | 62.8 | 53.8 |

43.0%. TUNERDEC-w/o-IR achieves the best performance among all approaches, with an average of 53.8% full match accuracy. This highlights the benefit of domain-specific tuning for structural recovery in code. Interestingly, TUNERDEC sees a slight decrease in CFG full match accuracy (42.1%) compared to its non-IR counterpart. This suggests a trade-off introduced by the iterative refinement process, which prioritizes functional correctness and re-executability over strict structural fidelity. Refinement often modifies control structures to pass test cases, potentially diverging from the original CFG. To address this trade-off, future work could explore more nuanced refinement strategies. For example, integrating a CFG-guided constraint into the refinement prompts may ensure the generated corrections remain structurally faithful.

## 8 RELATED WORK

Traditionally, decompilation relied on rule-based program analysis techniques to reconstruct high-level code from binaries. Tools like IDA Pro (Hex-Rays, 2024) and Ghidra (Agency, 2024) use hand-crafted rules to map machine code to language constructs such as functions, loops, and conditionals. However, developing rule-based decompilers requires years of manual effort (Katz *et al.*, 2019), and the resulting code is often difficult for humans to understand (Armengol-Estapé *et al.*, 2024).

Neural approaches, leveraging deep neural networks (DNNs), aim to automate decompiler construction by learning from data. For example, NeurDP (Cao *et al.*, 2022) trains a DNN on binary-source code pairs to predict high-level code structures, capturing complex patterns and generating more coherent outputs than traditional methods. Despite their potential, these models face challenges such as the need for large annotated datasets, which are labor-intensive to create, and poor generalization to unseen or highly optimized binaries, often resulting in syntactically correct but semantically flawed code (Liu and Wang, 2020).

Recent advancements have integrated LMs into decompilation tasks. SLADE (Armengol-Estapé *et al.*, 2024) and LLM4Decompile (Tan *et al.*, 2024) use transformers to generate high-level code from assembly or machine code. While promising, these approaches treat training data as plain text, failing to guide the model toward capturing high-level language constructs crucial for decompilation. Furthermore, iterative refinement techniques using LLMs have proven effective in automated code generation and refinement (Wong *et al.*, 2025; Cai *et al.*, 2025; Ding *et al.*, 2024; Madaan *et al.*, 2023). However, these methods focus mainly on generic tasks (e.g., competitive programming or simple debugging) and often overlook the structural semantics of low-level binary code. TUNERDEC overcomes these limitations via code-structure-aware fine-tuning and inference-time iterative refinement. Moreover, by integrating type prediction (Dramko *et al.*, 2025) and data-flow alignment (Feng *et al.*, 2024), TUNERDEC could achieve greater semantic and structural fidelity.

Our evaluation focuses on functions compiled with GCC on x86-64. This controlled setting ensures reproducibility and fair comparison with prior work. While it does not yet cover other architectures (e.g., ARM, RISC-V), future work will broaden evaluation to multiple architectures.

## 9 CONCLUSION

We have presented TUNERDEC, a neural-based approach for binary-to-C decompilation that integrates task-specific and general-purpose language models. Our proposed code-aware weighted token scheme, combined with an iterative refinement loop driven by compiler feedback, ensures high accuracy and structural integrity in the decompiled code. Evaluation results demonstrate that TUNERDEC outperforms existing decompilers in recompilation success, execution correctness, edit similarity, and CFG reconstruction performance to ground-truth C code. We used a large language model (ChatGPT 4/Grok3) solely for polishing grammar and improving the readability of the manuscript.

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
