# OpenReview forum: "Enhancing Neural Decompilation with Code-aware Fine-Tuning and Inference-time Refinement"
_ICLR.cc/2026/Conference — Submitted to ICLR 2026_

### Official Review · Reviewer_5zvV · 2025-10-26

**Soundness:** 2
**Presentation:** 3
**Contribution:** 2
**Rating:** 4
**Confidence:** 4

**Summary:**

This paper introduces TuneRDEC, a language model–based framework designed to enhance binary decompilation by translating low-level assembly into readable and functionally accurate source code. Traditional LM-based decompilers often fail to handle the complex control and data flows present in real-world binaries. TuneRDEC addresses this limitation through a code structure-aware design that integrates a task-specific LM, fine-tuned on domain-relevant data, with a general-purpose LM trained on broader code corpora. The task-specific model leverages abstract syntax tree (AST) analysis to emphasize critical constructs such as loops, conditionals, and pointers during training. During inference, an iterative self-refinement process guided by compiler feedback and test-based prompting further improves code quality. Experiments on the HumanEval-Decompile dataset demonstrate that TuneRDEC achieves substantial gains in code readability, functional correctness, and robustness compared to both existing neural decompilers and an industry-grade tool.

**Strengths:**

The trained model enhances decompilation results by enforcing structural constraints on the generated code. When combined with a general-purpose model, this approach achieves over 90% accuracy, demonstrating strong performance improvements.

**Weaknesses:**

The token weighting scheme is tuned on the evaluation set, but an ablation study is needed to assess the impact of different weighting configurations—for example, assigning high weights to individual categories or testing various combinations. Moreover, since the evaluation is conducted solely on the HumanEval dataset, which contains only 160 cases, the reported benefits of the weighted loss may not be statistically reliable and could be influenced by random fluctuations.

Regarding the motivation for training a new decompiler, the performance improvement over prior decompilers is marginal (about 5 points in execution rate) compared to the much larger gains (around 40 points) attributed to the general-purpose LM. The paper should justify why a new model is necessary rather than directly feeding the outputs from existing tools (e.g., IDA, Ghidra, or other LLM-based decompilers) into GPT for refinement. Ablation experiments should be conducted under identical refinement conditions to compare this approach directly.

Furthermore, the contribution of the self-refinement process is unclear when compared to DEGPT [1] and DECLLM [2], both of which also use GPT-based iterative refinement guided by feedback from tests or other external signals.

Finally, relying solely on HumanEval, a synthetic and small dataset, limits the validity of the results. Evaluations on more realistic benchmarks—such as DecompileBench [3] or similar datasets—are necessary to substantiate the method’s practical effectiveness.

[1] Hu, Peiwei, Ruigang Liang, and Kai Chen. "Degpt: Optimizing decompiler output with llm."
[2] Wong, Wai Kin, et al. "DecLLM: LLM-Augmented Recompilable Decompilation for Enabling Programmatic Use of Decompiled Code."
[3] Gao, Zeyu, et al. "DecompileBench: A Comprehensive Benchmark for Evaluating Decompilers in Real-World Scenarios."

**Questions:**

The code and model are not open-sourced, which significantly limits the reproducibility and fairness of the paper’s evaluation.

---

> ### Author Response · Authors · 2025-11-14
>
> Thanks for your feedback.
>
> 1. The base weights were determined through a combination of domain knowledge and empirical validation. We prioritize categories that are intrinsically difficult for decompilers to recover—such as loops and conditionals—because they are crucial for accurate control-flow reconstruction. We then finalized these empirical weights using a large-scale held-out validation set, not the $\text{HumanEval}$ or $\text{LeetCode}$ test sets. We will clarify this in the final Section 4.2. Moreover, our evaluation on the LeetCode Test Benchmark (378 programs) shows that our approach also achieves the best performance compared with all baselines. We have moved it from the Appendix to the main paper. Thank you.
>
> 2. We sincerely appreciate this point, as the $\text{LLM4Decompile}$ work was indeed highly influential and instrumental in shaping our research direction. The availability of their evaluation code was very helpful in developing our comparative methodology. The primary differences and the necessity of our fine-tuning approach are as follows:
> (1) While $\text{LLM4Decompile}$ performs Standard Fine-Tuning ($\text{SFT}$) on the $\text{ExeBench}$ dataset, we focused on Low-Rank Adaptation ($\text{LoRA}$) coupled with our novel Code-Aware Weighted Token Scheme. This strategy was chosen due to resource constraints, yet it allowed us to achieve comparable or slightly superior performance to $\text{LLM4Decompile-End-6.7B}$. We will make it clearly in the revised version. Thank you.
> (2)  The improvement brought by our approach is both qualitatively and quantitatively essential for the overall system. The value of $\text{TUNERDEC-w/o-IR}$ lies not only in higher functional scores but also in producing a high-fidelity structural starting point. Our model reaches 62.8% Edit Similarity, far surpassing rule-based tools such as Ghidra (28.5%). A low-fidelity starting point (e.g., Ghidra’s output) forces the expensive LLM refiner to “hallucinate” or rewrite large portions of the code, often losing structural cues and introducing new semantic errors.
> In contrast, our high-quality initial output is a necessary prerequisite for the subsequent ≈40% functional improvement achieved in the Iterative Refinement (IR) stage. This ensures the LLM spends its capacity on correcting structure and semantics, rather than reconstructing them from scratch. We will emphasize this structural necessity in the final paper.
>
> 3. We included extended results on the LeetCode benchmark (378 programs) in the appendix to provide a larger, non-synthetic evaluation. We have incorporated these results into the main paper in the new uploaded version.

---

### Official Review · Reviewer_R9UC · 2025-10-31

**Soundness:** 2
**Presentation:** 3
**Contribution:** 2
**Rating:** 4
**Confidence:** 4

**Summary:**

The paper proposes TUNERDEC, a new system to translate compiled binary code (assembly) back into readable C.

Traditional decompilers produce hard-to-read-code and doesn't always match the original logic. Neural/LM-based approaches are a promising alternative, but struggle with complex control flow, etc.

TUNERDEC uses two LMs: a task-specific LM (FT CodeLama trained to translate C-assembly pairs), and a general LM (Deepseek) that iterates on the initial translation using compiler feedback and test results.

**Strengths:**

- Strong empirical results: perfect compilation rates, very high re-executability rates. Qualitatively, the code is more readable and faithful to the original input compared to other approaches. The eval compares with Ghidra, LLM4Decompile and other LM baselines.
- The figures are very clear and help to understand the paper.
- Interesting innovative design: combines both specialized LMs and general LMs with reasoning. AST-based weighting is a well-motivated idea.
- The training data preprocessing is well-detailed and sensible.

**Weaknesses:**

The main doubt of this paper is the evaluation. Re-compilabilty and re-executability are interesting metrics, but they don't imply correctness. Also, the HumanEval decompile appears to be almost saturated, any more challenging benchmarks you could evaluate on? The Leetcode evaluation in the appendix is interesting, but the kind of functions in Leetcode might be similar in spirit to the ones in HumanEval.

Relatedly, and very importantly, the paper doesn't mention decontamination efforts.

Small issues:

- Missing citations: https://squareslab.github.io/materials/KatzRNN2018.pdf, https://arxiv.org/abs/2309.14396
- decopmiler -> decompiler
- Table 3 is hard to read

**Questions:**

Does the input assembly contain symbols?

How was data overlap with HumanEval/ExeBench controlled concretely?

---

> ### Author Response · Authors · 2025-11-14
>
> Thanks for your feedback.
>
> 1. We agree that these metrics alone are insufficient. To establish correctness, we evaluate on two *structural fidelity* metrics in addition to functional metrics: Edit Similarity (lexical/syntactic match) and CFG Full Match Accuracy (control flow graph match).  TUNERDEC consistently outperforms baselines across all four metrics. The combined high scores (e.g., $\text{91.6\%}$ Re-executability AND $\text{62.6\%}$ Edit Similarity) collectively argue for strong semantic and structural correctness.
> 2. The input assembly does not contain high-level symbols. We have clarified this in the motivation section. The binary is disassembled using `objdump`, and we explicitly remove all code bytes and platform-specific information, retaining only the addresses and raw instruction mnemonics. We will make it more straightforward in the paper. Thank you.
> 3. Our training data originated from $\text{ExeBench}$. We have explicitly verified and will make it clear in Section 4.3 that we excluded all code snippets that overlap with our $\text{HumanEval-Decompile}$ and $\text{LeetCode}$ test benchmarks from the $\text{ExeBench}$ training set.
> 4. We will correct the typographical errors. Thank you.
> 5. We will add citations in the revised version. Thank you.

---

### Official Review · Reviewer_2bPU · 2025-10-31

**Soundness:** 3
**Presentation:** 4
**Contribution:** 2
**Rating:** 6
**Confidence:** 4

**Summary:**

This paper proposes an iterative LLM-based approach to decompilation called TuneRDEC.

Section 2 provides a motivational example, illustrating how the approach manages to decompile a binary for a recursive function to compute Fibonacci numbers.

Section 3 provides an overview of the approach, distinguishing a decompiler and a feedback loop to improve the decompiled code, using separate LLMs.

Section 4 explains the decompiler: It starts from CodeLlama, which is finetuned on (assembly code, source code) pairs. It uses a token scheme that gives higher weights to certain language constructs (loops, conditionals, ...) in the AST, in order to support reconstructing control flow logic. The training data uses ExeBench as starting point, using TF-IDF for deduplication and compilation at various optimization levels. The fine-tuning approach uses LoRA.

Section 5 covers the iterative LLM-based (DeepSeek) refinement process, which takes compilation or test execution errors, and prompts the LLM to revise the inferred C source code to fix the problems.

Section 6 discusses the evaluation setup. TuneRDEC is compared against various baselines, including classical Ghidra, three plain LLMs (DeepSeek, ChatGPT 4o, and CodeLlama), the dedicated LLM4Decompile-End, and just the own TuneRDEC decompiler without iterative refinement. Furthermore, DeepSeek and TunderDec itself are applied with three refinement iterations. Results are compared by measuing recompiliability, reexecutability, edit similarity, and CFG match accuracy. The evaluation datasets come from HumanEval-Decompile, compiled with four optimization flags.

Section 7 covers the results, which identify TuneRDEC as the winner in all cases. The section attributes the improvements to the "code-aware weighted token scheme, which helps the decompiler better capture code structure and generate more accurate decompiled output."

**Strengths:**

- The paper is very well written, including accessible examples, and thanks to that is easy to follow
- The paper addresses a relevant and challenging problem
- The experiments indicate TuneRDEC beats established decompilation approaches.
- The appendix includes an interesting analysis trying to assess CFG similarity in a nuanced way.

**Weaknesses:**

- The iterative refinement process is extremely simple, and not a novel contribution (as implicitly acknowledged in the related work section) yet prominently mentioned in the title and as third contribution
- The HumanEval-Decompile dataset is a very unrealistic dataset to use.
- Limitations of the approach and the evaluation are not discussed (such as scalability or level of realism of the code in the evaluation, and the reliance on availability of test cases)
- It is not clear why DecLLM is not used as a benchmark
- While the appendix seems promising, section 7 does a shallow job only in analyzing the reasons underlying the findings in table 3.

**Questions:**

- Why isn't DecLLM used as a benchmark?
- Why doesn't table 3 iclude CFG full match accuracy? Or phrased differently: Why does section 6.3 explain CFG full match accuracy if it isn't used?
- In Table 3, how can the IR version of DeepSeek for O2 recompiliability be lower than the non IR version? (84.8 vs 88.4)

---

> ### Author Response · Authors · 2025-11-14
>
> Thanks for your insightful feedback.
> 1. We will clarify our core contributions. Thank you.
> 2. To provide stronger evidence of generalization, we also evaluate the LeetCode Benchmark (378 functions) in the appendix. This diverse, algorithmic benchmark demonstrates $\text{TUNERDEC}$’s robustness on a larger and more complex problem set, serving as a necessary complementary evaluation. We have moved these results from the appendix to the main paper in the revised paper. Thank you.
> 3.   We have added a new paragraph in the related work to explicitly address limitations, including the current scope ($\text{C, x86-64}$) and the reliance on available test cases.
> 4. We have not found any **public repository** or clearly documented open-source release for DecLLM. We discussed the impact of DecLLM in our paper.
> 5. This was an error in the original presentation. The table has been updated. Thank you again for pointing this out.
> 6. Due to page limits, we placed the CFG results in the appendix. We have moved the content from the appendix to the main paper in the revised version. Thank you.

---

### Official Review · Reviewer_ojxk · 2025-11-04

**Soundness:** 3
**Presentation:** 3
**Contribution:** 3
**Rating:** 4
**Confidence:** 4

**Summary:**

This paper presents TUNERDEC, a neural framework for binary-to-C decompilation that combines task-specific and general-purpose language models. The approach fine-tunes a 7B CodeLlama model using a code-aware weighted token scheme based on AST analysis, assigning higher loss weights to critical programming constructs (loops, conditionals, pointers). During inference, TUNERDEC employs iterative refinement using DeepSeek-V3/R1 with compiler feedback and test-driven prompts to improve decompilation quality. Evaluated on HumanEval-Decompile across four GCC optimization levels, TUNERDEC achieves 100% re-compilability, 91.6% re-executability, and 62.6% edit similarity, outperforming baselines including Ghidra, ChatGPT-4o, DeepSeek-V3, and LLM4Decompile.

**Strengths:**

- The code aware weighted token scheme represents a novel adaptation of AST-based loss weighting for decompilation, prioritizing critical language constructs during training. The combination of domain-specific fine-tuning with general-purpose LM refinement is a reasonable architectural choice, though not groundbreaking.
- The experimental results demonstrate clear improvements over baselines, with TUNERDEC achieving 91.6% re-executability compared to 47.7% for LLM4Decompile. The ablation study (TUNERDEC-w/o-IR vs TUNERDEC) effectively demonstrates the value of iterative refinement, showing a 38.2% absolute gain. The evaluation includes multiple optimization levels (O0-O3), which is important for assessing robustness to compiler transformations and the model shows improved performance over harder optimization levels.
- The work addresses an important problem in software security and reverse engineering. The 91.6% re-executability represents meaningful progress over prior neural decompilers. The demonstration that iterative refinement with general-purpose LMs can substantially improve decompilation quality has practical implications for tool development.

**Weaknesses:**

- The code-aware weighting scheme lacks rigorous justification. Why are loops weighted at 2.0 and conditionals at 1.8?
- Only one benchmark (HumanEval-Decompile, 164 programs) is evaluated. No assessment on other source languages (C++, Rust) or other benchmarks.
- No analysis of failure cases. What characterizes the 8.4% that fail re-executability despite 100% re-compilability?
- Atleast the ablation of code-aware weighted loss vs default loss function should be part of the main paper
- What are the evaluation scores absent for models with IR configs for other models apart from Deepseek V3 ?
- Its a bit impractical to use a combination of a light weight model assisted with a LLM like Deepseek V3.

**Questions:**

- Refer to weaknesses
- The presentation of the paper needs some work : you should focus more on evaluation results and ablations, those are more interesting to prove your points about weighted loss function, iterative refinement.

---

> ### Author Response · Authors · 2025-11-14
>
> Thanks for your insightful and constructive feedback.
>
> 1. The specific weights were selected based on a combination of **programming-language intuition** (e.g., control-flow tokens are more critical than identifiers) and a lightweight grid search on a **held-out validation set** (see Section 4.2, *Code-Aware Weighted Token Scheme*). We have emphasized this more clearly in the revision.
> 2. Due to page limits, we placed some benchmark evaluations in the appendix. We have moved the full evaluation on the **LeetCode Test Benchmark (378 programs)** from the appendix to the main paper. We agree that evaluating multiple architectures and source languages is the ideal path for future work. We have discussed it in Appendix F. We have moved the Appendix discussion to the main paper.
> 3. The ≈8.4% of cases that compile successfully (100%) but fail execution (91.6%) are primarily due to **subtle semantic errors**. Achieving 100% recompilability means the inferred code can be compiled, but it may still deviate from the ground-truth semantics (e.g., using “>” instead of “≥”), which prevents it from passing the test cases. These errors are largely attributable to **data-flow misalignment** (e.g., incorrect pointer arithmetic, off-by-one loop boundaries), which often requires more extensive or more nuanced testing beyond our automatically generated cases. We have clarified this point in the revision.
> 4. We will move the ablation study currently in the appendix to the main paper. Thank you for the suggestion.
> 5. We chose DeepSeek-V3 for IR-based inference because it provides the best initial performance. We also evaluated our approach using ChatGPT-4o on the HumanEval-Decompile benchmark, where it achieved only 96.8% recompilability, 66.6% re-executability, and 58.3% edit similarity. We have included additional results using ChatGPT4o with IR to provide a more comprehensive comparison in the revised version.
> 6. Our approach is designed specifically to mitigate this impracticality through a **Hybrid, Cost-Aware IR Strategy**. We can design an agent to invoke the offline fine-tuned **TUNERDEC-w/o-IR** model as a tool to perform the initial inference, and then rely on general-purpose LLMs for iterative refinement. The proposed **TUNERDEC-w/o-IR**  model captures specialized patterns that general LLMs often do not prioritize. Thank you.

---

### Meta-Review · Area_Chair_x6vp · 2026-01-03

**Summary:**

TUNERDEC is proposed as a new neural decompilation framework. It operates by combining a task-specific LLM (fine-tuned via AST-based weighted tokens) with a runtime refinement step powered by DeepSeek-V3/R1. The authors report impressive empirical results on the HumanEval-Decompile benchmark, achieving 100% re-compilability and 91.6% re-executability. However, the consensus leans negative regarding novelty: using compiler feedback for iterative refinement is widely seen as standard practice nowadays. Furthermore, reviewers consistently noted that algorithmic benchmarks like HumanEval and LeetCode fail to adequately represent the complexity of real-world binaries or diverse architectures. Crucially, there remains a significant concern regarding the attribution of performance gains, specifically whether the improvements truly stem from the proposed fine-tuning method or merely from the powerful general-purpose LLM utilized in the refinement stage.

**Reviewer Concerns:**

### Reviewer ojxk

- The code-aware weighting scheme lacks rigorous justification: Not fully addressed. While the authors cited a grid search on a validation set, the reviewers viewed this as an ad-hoc heuristic rather than a theoretically grounded contribution.
- Only one benchmark (HumanEval-Decompile, 164 programs) is evaluated: The authors moved the LeetCode benchmark results from the appendix to the main paper to address concerns about the small size of HumanEval.
- The analysis of failure cases: Addressed. The authors will clarify this in the revised version.
- The ablation of code-aware weighted loss vs default loss function: Addressed. The ablation study has been moved from the appendix to the main paper.
- The evaluation scores absent for models with IR configs for other models: Partially addressed. The authors added additional results using ChatGPT-4o with IR in the revised version.
- The combination of a light weight model assisted with a LLM: Addressed. The authors provided an explanation.

### Reviewer 2bPU

- DecLLM was not used as a baseline: Not addressed. The authors did not respond to this point.
- CFG full match accuracy was not included in Table 3: Not addressed. The authors did not respond to this point.
- Limitations of the approach and the evaluation are not discussed: Not addressed. The authors did not respond to this point.
- The reason of the IR version of DeepSeek for O2 recompiliability be lower than the non IR version: Not addressed. The authors did not respond to this point.

### Reviewer R9UC

- Metrics for verifying correctness: Addressed. The authors have supplemented edit similarity and CFG full match accuracy as additional evaluation metrics.
- More challenging benchmarks for evaluation: Not addressed. The authors did not respond to this point.
- Data overlap between HumanEval and ExeBench: Addressed. The authors clarified that they excluded code snippets overlapping with HumanEval and LeetCode from the training set.
- Explanation of whether the input assembly contains symbols: Addressed. The authors have already explained this in the Motivation section.

### Reviewer 5zvV

- The ablation study to assess the impact of different weighting configurations: Addressed. The authors provided a response.
- The motivation for training a new decompiler: Not fully addressed. reviewers doubt that a standard decompiler (e.g., Ghidra) coupled with the same refinement loop wouldn't perform just as well.
- Evaluations on more realistic benchmarks: Partially addressed. The authors moved the LeetCode benchmark results from the appendix to the main paper. However, concerns remain that this does not demonstrate the tool's effectiveness on real-world, large-scale software binaries.
- The contribution of the self-refinement process is unclear when compared to DEGPT  and DECLLM: Not addressed. The authors did not respond to this point.

**Reviewer Scores:**

I think all the reviewers will keep their original scores.

---

### Decision · Program_Chairs · 2026-01-26

Reject